# SGD Smooths the Sharpest Directions

**Stanisław Jastrzębski**[1,2]**, Zachary Kenton**[2]**, Nicolas Ballas**[3]**, Asja Fischer**[4]**,
Yoshua Bengio**[2] **& Amos Storkey**[5]

[1]Jagiellonian University, Cracow, Poland
[2] MILA, Université de Montréal, Canada
[3] Facebook, Montreal, Canada
[4] University of Bonn, Bonn, Germany
[5] The University of Edinburgh, United Kingdom

## Abstract

Stochastic gradient descent (SGD) is able to find regions that generalize well, even in drastically over-parametrized models such as deep neural networks. We observe that noise in SGD controls the spectral norm and conditioning of the Hessian throughout the training. We hypothesize the cause of this phenomenon is due to the dynamics of neurons saturating their non-linearity along the largest curvature directions, thus leading to improved conditioning.

## 1 Introduction

Deep neural networks (DNNs) are massively overparameterized models (Zhang et al., 2016), yet they show state-of-art generalization performance on a wide variety of tasks when trained with stochastic gradient descent (SGD). While understanding the generalization capability of DNNs remains an open challenge, it has been hypothesized that SGD acts as an implicit regularizer, limiting the complexity of the found solution (Advani & Saxe, 2017; Jastrzębski et al., 2017; Poggio et al., 2017; Wilson et al., 2017). The dynamics of how this occurs remain unclear.

It is a shared intuition that high noise in SGD smoothes out the loss surface (Bottou, 1991). Modifications to SGD aimed at achieving a smoother loss have been proposed in, for example, (Chaudhari et al., 2016; Gülçehre et al., 2016). Our main observation is that SGD implicitly smoothes out the loss surface, by escaping sharp regions of the loss surface.

## 2 SGD smoothing effect in Neural Networks

Our goal is to empirically study the impact of high noise in SGD on the geometry of the DNN loss surface explored by SGD. We perform experiments on multilayer perceptrons (MLP) and convolutional neural networks. MLP experiments are run on a teacher-student task of (Advani & Saxe, 2017), where a teacher network implements a noisy mapping from inputs to labels with weights sampled from a Gaussian of unit variance, and noise sampled from a Gaussian of variance 0.2. The input data has dimension $d = 300$ and is sampled from a Gaussian of variance $1/300$. The student network tries to learn the teacher weights. MLP have two layers of hidden size 100, and use the rectifier linear non-linearity.

CNN experiments are run on CIFAR10 and CIFAR100 (Krizhevsky et al.) datasets using the VGG11 (Simonyan & Zisserman, 2014) and residual network (He et al., 2015) models. Unless specified otherwise, models are trained using SGD with $S = 128$, $\eta = 0.1$ and momentum 0.9. Computing the full spectrum in such a model is computationally infeasible. We approximate the top $K$ eigenvalues (in absolute) using the Lanczos algorithm (Lanczos, 1950; Dauphin et al., 2014), an extension of the power method, on approximately $5\%$ of the training data. We do not distinguish in the CNN experiments between negative and positive eigenvalues. The spectrum is computed with regularization (dropout and batch normalization) switched off. We use $L2 = 5 \cdot 10^{-4}$ weight decay for both models, and dropout 0.5 in the classifier of VGG11. Experiments are performed using Keras (Chollet et al., 2015) and Tensorflow (Abadi et al., 2015).

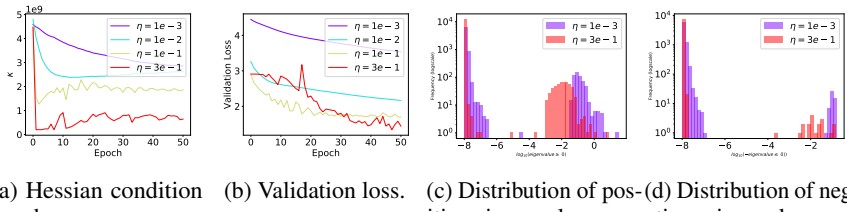

(a) Hessian condition number.  (b) Validation loss.  (c) Distribution of positive eigensvalues.  (d) Distribution of negatives eigenvalues.

Figure 1: Two-layers MLP trained on the teacher-student task.

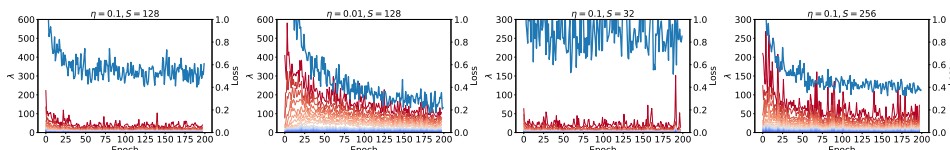

Figure 2: The top 30 eigenvalues of the Hessian (colored red to blue) of Resnet32 on CIFAR10 and loss on the training set (thick blue line). We observe an oscillation in the top eigenvalues and in the loss curve. Additionally, for same loss value lower $\eta$ or higher $S$ steers to Hessian with higher spectrum. Top row compares $\eta = 0.1$ with $\eta = 0.01$. Bottom row compares $S = 32$ with $S = 256$.

We first explore the impact of high learning rate by looking at the spectrum of the Hessian of our model, through training. To ease the computational burden associated with the Hessian spectrum computation, we consider a small two-layer MLP model and the teacher-student task. We train several MLPs using SGD for 50 epochs with different learning rates.

We report the Hessian condition number $\kappa(H)$ throughout training in Fig. 1(a). The model trained with higher learning rate, reaches lower $\kappa(H)$. Next, we look at the full Hessian spectrum for a MLP trained with high learning rate and low learning rate in Fig. 1(c) and (d). We confirm the finding of Sagun et al. (2016; 2017) observing that the the Hessian is ill-conditioned for both high-noise and low-noise model. Further, we see that the high learning rate reduces the magnitude of the large eigenvalues (in absolute). We also report that low $\kappa(H)$ correlates with better performance on unseen data in Fig. 1 (b).

Next, we confirm that noise in SGD has similar effects in deeper convolutional models. We track the top 30 (in absolute value) eigenvalues of the Hessian using the Lanczos approximation algorithm for Resnet32 trained using different learning rates and batch-sizes on CIFAR10. In Fig. 2, top, we report how the eigenvalues evolve through the training epochs, for different learning rates. Higher learning rate leads to generally lower spectral norm, even for the same loss value. We also notice that this is especially prominent early in the training.

To ensure robustness of our results, for Resnet32 and VGG11 on CIFAR10 we run a grid of 11 values of $\eta$ in the range $[10^{-4}, 10^{-1}]$. Each model is trained for 200 epochs and we measure the average (over the course of training) of the top $k$ eigenvalues spectrum and of the spectral norm. Fig. 3 (left) reports a summary of the considered grid. We observe an approximately linear relation between $\frac{1}{\eta}$

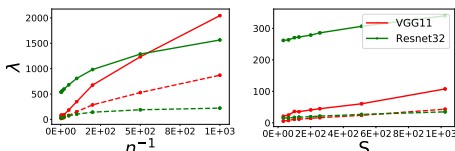 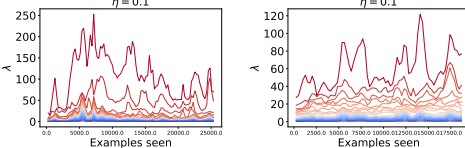

Figure 3: Hyperparameters of SGD control the spectrum of Hessian. The larger $1/\eta$ (left), or $S$ (right) in SGD the larger the average of the top 10 eigenvalues (dotted line), and spectral norm (solid line).

Figure 4: Zoom on the first epoch of training Resnet32 (left) and VGG (right) shows high frequency (batch scale) oscillatory evolution of the top eigenvalues of the Hessian.

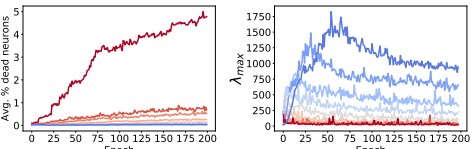 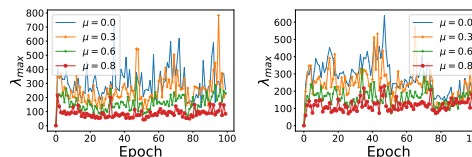

Figure 5: Evolution of average percentage of dead neurons (left) compared with evolution of the spectral norm (right) for different $\eta$ for Resnet32. Learning rate from highest to lowest in red to blue.

Figure 6: Momentum reduces spectral norm of the Hessian. Training at $\eta = 0.1$ and $S = 128$ on VGG11 and CIFAR10 (left) and CIFAR100 (right).

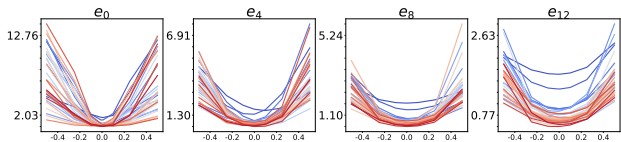

Figure 7: Slice of the loss surface along eigenvectors (columns) for Resnet32 with $\eta = 0.1$ in the first 100 epochs. Each subplot represents time evolution of the loss surface along one eigenvector (y axis is loss, x axis is distance) over the first 100 batches of Resnet32 training (blue to red).

and each of spectral norm and average of the top 10 eigenvalues. Next, we run a similar experiment changing $S$ in range $[32, 1024]$, and observe a similar linear relation, see Fig. 3 (right).

Next, we zoom onto first epoch of training Resnet32. We observe that high noise in SGD leads to high frequency oscillations in the spectral norm, see Fig. 4. To further examine the phenomena, in Fig. 7, for each eigendirection, we plot how the loss surface changes in this first epoch along the eigendirections. We use a constant step along the eigenvectors scaled by $\eta$.

Finally, we explore the impact of momentum, observing that large momentum also leads to a a drastic reduction of spectral norm of the Hessian, see Fig. 6 for results on the VGG11 network on CIFAR10 and CIFAR100.

## 2.1 WHY HIGH NOISE REDUCES THE SPECTRAL NORM

In linear regression, the behaviour along a given eigenvector depends on the product of $\eta$ and its eigenvalue $\lambda$. If the product is too high then, this direction will diverge. However, when training DNNs, SGD is able to escape regions that lead to such divergence. Figs. 2, 4 and 7 report that the magnitude of the largest eigenvalues oscillate through training of Resnet32 with $\eta = 0.1$, and hence $\rho(H)$ oscillates. To explain the cause of this phenomena, we hypothesize that SGD takes too large a step along the largest curvature directions which leads to an increase in the percentage of dead neurons due to saturation, which in turn leads to decrease of the eigenvalue associated with the largest curvature directions.

We consider that a neuron is saturated or dead if the mean absolute activation over a batch of examples is less than $10^{-3}$. In Fig. 5 we report the number of dead neurons through training of Resnet32 on CIFAR10 as well as the Hessian spectral norm. Higher learning rate corresponds both to a higher percentage of dead neurons and lower Hessian spectral norm.

## 3 CONCLUSIONS

We investigated how noise in SGD dynamically impacts the loss geometry throughout training. We observe that noise in SGD controls the magnitude of the top eigenvalues (in absolute) of the Hessian throughout training. We observe that SGD with larger noise reduces the top eigenvalues. This behaviour acts as an implicit regularizer as it leads to a better conditioning of the Hessian which is known to correlate with better generalization performances.

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
