# OpenReview forum: "SGD Smooths The Sharpest Directions"
_ICLR.cc/2018/Workshop — Reject_

### Official Review · AnonReviewer1 · 2018-03-01
**SGD noises help training**

**Rating:** 6
**Confidence:** 3

**Review:**

This paper studies how the magnitude of SGD noises (parameterized via the learning rate) affects the spectrum and condition number of the Hessian, as well as the rate of dead neurons. The experiments are interesting, though more analysis might be better. Also, the reviewer would like to see the validation error with respect to the SGD noises on the classification tasks, too. In particular, it would be good to also show Figure 1(b) for the classification tasks. Besides, some of the experiments could also be refined for more controlled comparison. For example, comparing different learning rates with the same number of epochs might not be perfectly fair as smaller learning rate learners move less far than larger learning rate learners in the parameter space.

---

### Official Review · AnonReviewer3 · 2018-03-06
**Hard to read and feels like sections are missing**

**Rating:** 3
**Confidence:** 4

**Review:**

This is an empirical paper that tries to prove that (i guess) SGD serves as a some sort of the regularizer  due to the level of noise. I feel it would have been interesting, but it is so poorly organized and presented that the whole story is lost.

Overall the paper feels rushed and is missing sections that could facilitate the reading. Introduction is rushed. Furthermore, the main section that would explain what the authors were really trying to prove is missing - the paper jumps directly to the experiments, leaving the reader trying to guess what is that they are testing for and how they go about it.
What are you trying to show? How are you planning on showing it and why it is the right way of doing it?
A couple of lines like 'We look at the hessian condition number to see how susceptible the network to noise" would have went a long way to facilitate the reading

The first set of experiments is dealing with learning rate. But is it the only way to regulate the noise? Will increasing the batch size reduce the noise? if it is the case, then you would expect that generalization will suffer when larger batches are used? Why not to show those experiments?

On top of that, for an empirical paper, showing it just on 3 datasets is not enough. How were the hyperparameters chosen. Were they tuned? How? Can it be that the observations hold only for those values of hyperparams

Minor: The abstract is stated as the fact, instead of We show .. etc

---

### Official Review · AnonReviewer2 · 2018-03-12
**A work in progress**

**Rating:** 5
**Confidence:** 4

**Review:**

This paper empirically shows that learning rate and batch size in SGD influence the spectral norm of the Hessian of the weights. Although I am not aware of these specific experiments having been done in other works, the idea that higher learning rates avoid sharp minima is widely known (i.e. gradient descent's optimal learning rate is equal to the inverse of the second derivative).

There are some issues with the paper:
- I don't think that reporting the condition number of the Hessian in Figure 1a is meaningful, as this is sensitive to the magnitude of the smallest eigenvalue which can be unreliable due to rounding errors. It's hard to draw conclusions based on a difference between a condition number of 1e9 and 5e9 - this could be due to a difference between smallest eigenvalues of magnitude 1e-5 and 2e-6. Is the difference between those two tiny eigenvalues really meaningful?
- Figure 2 is confusing - the eigenvalues are colored red to blue, but the blue line also indicates the training loss. It would help to use a different color altogether for the training loss.
- Figure 7 is hard to interpret and doesn't have an explanation, please discuss a bit more what these plots are supposed to show.
- I'm not convinced by Section 2.1 and Figure 5. First, the title says "High Noise reduces the spectral norm", but the plots are with respect to learning rates, not noise. Second, the decrease in spectral norm with higher learning rates could simply be explained by gradient descent not being able to settle in sharp valleys when the learning rate it too high, and preferring wide basins (see for example: http://srdas.github.io/DLBook/GradientDescentTechniques.html).
- x labels in Figure 1c, d are too small

This paper is still definitely a work in progress, it could still make an acceptable workshop contribution.

---

### Decision · Program_Chairs · 2018-03-20
**ICLR 2018 Workshop Acceptance Decision**

**Decision:**

Reject

**Comment:**

Based on the reviews, this paper has not been accepted for presentation at the ICLR workshop. However, the conversation and updates can continue to appear here on OpenReview.